# Synthesis, Structure and Properties of Polyester Polyureas via a Non-Isocyanate Route with Good Combined Properties

**DOI:** 10.3390/polym16070993

**Published:** 2024-04-04

**Authors:** Liuchun Zheng, Qiqi Xie, Guangjun Hu, Bing Wang, Danqing Song, Yunchuan Zhang, Yi Liu

**Affiliations:** 1School of Chemical Engineering and Technology, Xinjiang University, Urumqi 830017, China; 2School of Chemical Engineering and Technology, State Key Laboratory of Separation Membranes and Membrane Processes, Education Ministry Key Laboratory of Hollow Fiber Membrane Materials and Membrane Processes, Tiangong University, Tianjin 300387, China; 3Cangzhou Insititute of Tiangong University, Cangzhou 061000, China; 4Shenghong Advanced Materials Research Institute, Shanghai 201403, China

**Keywords:** non-isocyanate, polyurea, polyester, soft segment, impact resistance

## Abstract

Polyureas have been widely applied in many fields, such as coatings, fibers, foams and dielectric materials. Traditionally, polyureas are prepared from isocyanates, which are highly toxic and harmful to humans and the environment. Synthesis of polyureas via non-isocyanate routes is green, environmentally friendly and sustainable. However, the application of non-isocyanate polyureas is quite restrained due to their brittleness as the result of the lack of a soft segment in their molecular blocks. To address this issue, we have prepared polyester polyureas via an isocyanate-free route and introduced polyester-based soft segments to improve their toughness and endow high impact resistance to the polyureas. In this paper, the soft segments of polyureas were synthesized by the esterification and polycondensation of dodecanedioic acid and 1,4-butanediol. Hard segments of polyureas were synthesized by melt polycondensation of urea and 1,10-diaminodecane without a catalyst or high pressure. A series of polyester polyureas were synthesized by the polycondensation of the soft and hard segments. These synthesized polyester-type polyureas exhibit excellent mechanical and thermal properties. Therefore, they have high potential to substitute traditional polyureas.

## 1. Introduction

Polyurea is a kind of high-performance polymer material consisting of a urea group repeating unit with symmetric double-dentate hydrogen bonds [1]. Compared with single hydrogen-bonded polyamides and polyurethanes, double-dentate hydrogen bonds endow polyurea with a high crystallinity, melting point, rigidity and polarity [2]. As a result, polyurea possesses excellent tensile strength, heat resistance, bending resistance, corrosion resistance, solvent resistance and friction resistance. Therefore, polyurea has been widely applied in many fields, such as fibers, membranes, coatings and so on. In 2022, the global annual market value of polyurea reached $885 million, and it is expected to reach $1481 million by 2025 [3]. Polyureas are traditionally produced with diisocyanate (or polyisocyanate) and incorporate polyether or diamine as a chain extender. Isocyanates and their precursor, phosgene, are highly toxic and harmful to human health and the environment [4]. As isocyanate is also highly reactive, it is very hard to preserve and control its reaction [5]. In addition, it is a significant challenge to process polyureas because the melting temperature of polyureas with isocyanates is close to their decomposition temperature due to strong intermolecular hydrogen bonding [6].

To address the above issue, various methods have been tried to synthesize polyureas by non-isocyanate methods. Generally, non-isocyanate polyureas can be synthesized by carbon dioxide/diamine [7,8,9,10,11,12,13], carbamate/diamine [14,15,16,17,18,19,20,21] and urea/diamine reactions [1,5,22,23,24,25,26]. Shi et al. prepared polyureas with excellent mechanical and thermal properties from carbon dioxide and diamines by a two-step process [8], but the reaction requires harsh conditions of high pressure, reaching as high as 11.5 MPa, which is potentially dangerous. Synthesis of polyureas from carbonate with diamines has attracted extensive attention. Kébir et al. synthesized polyureas with molecular weights (*M*_n_) ranging from 1.7 × 10^3^ to 2.7 × 10^3^ using dimethyl carbamates and various diamines as raw materials and 1,5,7-triazabicyclo [4.4.0]dec-5-ene as catalyst [14]. As the *M*_n_ is relatively low, the processing properties and mechanical properties of the polyureas are poor, and their application is greatly restrained. Li et al. synthesized thermoplastic polyureas with excellent mechanical properties from diethylene glycol bis(3-aminopropyl) ether, bis(hydroxyethyl) hexanediurethane and bis(hydroxyethyl) isophoronediurethane by melt polycondensation [16]. However, the preparation process of intermediate product of bis(2-hydroxyethyl) diaminoate and dimethyl carbamates is complex and requires repeated purification, and the yield is relatively low. In contrast, the reaction conditions of polyureas from urea are mild, and do not require high pressure. Lin et al. produced biobased polyurea (PUr) from urea, biobased 1,6-diaminohexane (DA-6) and 1,10-diaminodecane (DA-10) [5]. The mechanical properties of PUr can be regulated by the feed ratio of DA-6 to DA-10. The results showed that PUr-6_67_10_33_ exhibited a high tensile strength of 61 MPa and good elongation at break of 420%.

However, the chemical structure and properties of polyurea prepared by non-isocyanate methods differ significantly from those of polyurea prepared from isocyanate methods, which usually consists of two components, A and B. Component A is isocyanate or its prepolymer, and component B is the mixture of polyether amines, chain extenders and additives. The reaction of components A and B generates polyurea as a hard segment and polyether as a soft segment to endow the polymer with excellent mechanical properties [27]. The hard segments impart high modulus, strength and hardness to polyureas [28], while the soft ones provide toughness and impact resistance to polyureas [29]. Unfortunately, polyureas synthesized from non-isocyanate routes often lack soft segments. Consequently, they usually have shortcomings such as poor toughness and low impact resistance.

In this work, for the first time, we synthesized non-isocyanate polyureas with polyester as a soft segment. Firstly, urea and DA-10 were employed to synthesize polyureas prepolymer (Pre-PU) as the hard segment. Then, the soft segment, pre-polymerized poly(butylene dodecanedioate) (Pre-PBD), was synthesized by esterification and polycondensation of dodecanedioic acid (DDCA) and 1,4-butanediol (BDO). Finally, the polyester polyurea was synthesized through melt polycondensation of the soft segment and hard segment. The introduction of a soft segment largely improves the toughness and impact resistance of polyurea and expands its potential applications.

## 2. Materials and Methods

### 2.1. Materials

Urea (99.5%) and DA-10 (97%) were purchased from Macklin (Tianjing, China). DDCA with a purity of 99% was bought from Aladdin Biochemical Technology Co., Ltd. (Shanghai, China). The 1,4-butanediol (BDO) (99% purity) and stannous octoate (Sn(Oct)_2_, >96.0%) were purchased from J&K Chemical (Beijing, China). All the raw materials were used without further purification before the reaction.

### 2.2. Synthesis of Polyureas (PBD_x_PU_y_)

Polyurea was synthesize by the polycondensation of hard and soft segments. Pre-PU and Pre-PBD, corresponding to the hard and soft segments, were first synthesized, and then polycondensation was conducted to prepare a series of PBD_x_PU_y_ with different mass ratios of Pre-PU to Pre-PBD. The polymerization and polycondensation processes are described as below.

#### 2.2.1. Synthesis of Pre-PU

Pre-PU was synthesized via a three-step polycondensation method. Urea (1.6 mol, 96.10 g) and DA-10 (1.76 mol, 303.26 g) were fed to a 1 L kettle equipped with a nitrogen (N_2_) inlet and a fractionating column. The reactants were maintained at 130 °C for 2 h. Then, the temperature was raised to 210 °C and maintained for 1.5 h. Finally, the temperature was raised to 240 °C at 80 Pa for 20 min to obtain the Pre-PU.

^1^H NMR data (400 MHz, C_2_DF_3_O_2_-d_1_, δ in ppm): 3.35-3.24 (a, -C**H**_2_NH-), 3.24-3.15 (f, -C**H**_2_NH_2_), 1.79-1.68 (g, -C**H**_2_CH_2_NH-), 1.67-1.54 (h, -C**H**_2_CH_2_NH-), 1.44-1.21 (c, d, e, -C**H**_2_C**H**_2_C**H**_2_CH_2_CH_2_NH-).

#### 2.2.2. Synthesis of Pre-PBD

Pre-PBD was synthesized through a two-step reaction involving esterification and polycondensation. DDCA (1.2 mol, 272.36 g) and BDO (1.44 mol, 129.60 g) were fed into a 1 L kettle. The reaction system was heated to 160 °C under stirring with the protection of N_2_. Then, Sn(Oct)_2_ (0.0804 g, 0.02 wt%) was added as a catalyst. The esterification then proceeded until the esterification yield reached 90%. Then, the temperature was raised to 230 °C and the system pressure was reduced to 80 Pa for 0.8 h. The product was cooled by water and cut into pellets.

^1^H NMR data (400 MHz, C_2_DF_3_O_2_-d_1_, δ in ppm): 4.23-4.10 (a, -CH_2_C(O)OC**H**_2_-), 2.42-2.30 (b, -C**H**_2_C(O)OCH_2_-), 1.89-1.81 (c, -CH_2_C**H**_2_CH_2_OH), 1.80-1.67 (d, -CH_2_C(O)OCH_2_ C**H**_2_-), 1.67-1.50 (e, -C**H**_2_CH_2_C(O)OCH_2_-), 1.35-1.14 (f, -C**H**_2_C**H**_2_C**H**_2_CH_2_CH_2_C(O)OCH_2_-).

^1^H NMR data (400 MHz, CDCl_3_-d_1_, δ in ppm): 4.12-4.04 (a, -CH_2_C(O)OC**H**_2_-), 3.69-3.65(b, -C**H**_2_OH), 2.34-2.21 (c, -CH_2_C**H**_2_C(O)OCH_2_-), 1.71-1.65 (d, -CH_2_C(O)OCH_2_C**H**_2_-), 1.64-1.55 (e, -C**H**_2_CH_2_C(O)OCH_2_-), 1.34-1.22 (f, -C**H**_2_C**H**_2_C**H**_2_CH_2_CH_2_C(O)O-).

#### 2.2.3. Synthesis of PBD_x_PU_y_

A typical procedure for preparing PBD_x_PU_y_ is described below: predetermined amounts of Pre-PBD and Pre-PU were added to a kettle equipped with a nitrogen inlet and a fractionating column and stirred at 230 °C under the pressure of 80 Pa for 1–3 h. In order to obtain polymers with similar molecular weight, the polycondensation process continued until the torque reach a fixed value (5.5 N·m). For PBD_x_PU_y_, x and y represent the mass fraction of Pre-PBD and Pre-PU, respectively. For example, PBD_70%_PU_30%_ means the sample whose mass ratios of Pre-PBD and Pre-PU are 70% and 30%, respectively. The prepared samples are named as PBD_80%_PU_20%_, PBD_70%_PU_30%_, PBD_60%_PU_40%_ and PBD_50%_PU_50%_.

^1^H NMR data (400 MHz, C_2_DF_3_O_2_-d_1_, δ in ppm): 4.28-4.10 (a, -CH_2_C(O)OC**H**_2_-), 3.53-3.42 (d, -CH_2_NHC(O)C**H**_2_-), 3.36-3.18 (c, -C**H**_2_NHC(O)NHC**H**_2_-), 2.71-2.59 (b, -C**H**_2_NHC(O) CH_2_-), 2.48-2.28 (e, -C**H**_2_C(O)OCH_2_-), 1.82-1.67 (f, -C**H**_2_CH_2_O(O)CCH_2_CH_2_-), 1.62-1.52 (g, -CH_2_C**H**_2_CH_2_C(O)OCH_2_CH_2_CH_2_- and -C**H**_2_CH_2_NHC(O)NHCH_2_C**H**_2_-), 1.40-1.12 (h, -C**H**_2_ C**H**_2_C**H**_2_CH_2_CH_2_C(O)OCH_2_- and -C**H**_2_C**H**_2_C**H**_2_CH_2_CH_2_NHC(O)NH-).

### 2.3. Characterizations

The synthesized Pre-PBD, Pre-PU and polyurea were characterized by ^1^H NMR and FT-IR spectroscopy. ^1^H-NMR spectra were obtained using a Bruker A VIII600 NMR spectrometer at 25 °C (500 MHz, Bruker BioSpin Co., Ettlingen, Germany). The solvents were deuterated chloroform (CDCl_3_) or deuterated trifluoroacetic acid (C_2_DF_3_O_2_). An FTIR spectrometer (Varian 640-IR, Varian, Sydney, Australia) was used to test the FTIR spectra of samples, and the scan range was 4000 to 400 cm^−1^.

Thermal properties were measured using differential scanning calorimetry (DSC) (DSC-Q5800, Perkin-Elmer Co., Waltham, MA, USA). A sample of approximately 5–10 mg was placed in an aluminum crucible and scanned from 50 °C to 250 °C at 10 °C/min and kept there for 10 min under high-purity N_2_ to remove thermal history. Then, the sample was cooled to −25 °C at 100 °C/min and kept there for 10 min, and subsequently reheated to 250 °C at a rate of 10 °C/min. After maintaining at 250 °C for 5 min, the sample was then cooled to −25 °C at a rate of 10 °C/min. Both heating and cooling curves were recorded. The thermal stability of PBD_x_PU_y_ was tested on a thermogravimetric analyser (TGA) (NETZSCH TG 209 F3, Selb, Germany). The samples were heated at a rate of 20 °C/min from 50 °C to 800 °C under an N_2_ atmosphere. Dynamic mechanical analysis (DMA) of PBD_x_PU_y_ was performed using a DMA242E (Netzsch Co., Selb, Germany). It was conducted in shear mode, using a temperature sweeping from −90 °C to 40 °C, with a heating rate of 5 °C/min.

Wide angle X-ray analysis (WAXD) was conducted using an X-ray diffractometer (Bruker D8 DISCOVER, German Bruker company, Karlsruhe, Germany). The samples were scanned from 7° to 45° at a scanning speed of 2°/min using a Cu Kα radiation (λ = 154 nm) target.

In order to explore the mechanical properties of PBD_x_PU_y_, PBD and PU, tensile properties, flexural properties and impact properties were studied. The tensile testing of dumbbell-shaped samples (80.0 mm × 5.0 mm × 2.0 mm) was performed on an Instron 1122 tensile tester according to ISO 527 [30]. The crosshead speed was 50 mm/min. The flexural testing of samples (80.0 mm × 10.0 mm × 4.0 mm) was conducted on the same equipment according to ISO 178 [31], and the bending modulus was calculated from the slope of the stress-strain curve in the strain range of 0~1%. V-Notch impact strength testing of samples (80.0 mm × 10.0 mm × 4.0 mm) was performed on a pendulum impact tester (Shenzhen Sansi Co., Shenzhen, China) according to ISO 179 [32]. At least five specimens were tested for each sample, and the average value was reported.

The fracture surfaces of samples were coated with gold. Then, they were characterized using a scanning electron microscope (MERLIN Compact, Zeiss, Oberkochen, Germany) with an accelerating voltage of 15 kV.

## 3. Results and Discussion

### 3.1. Chemical Structure Characterization of PBD_x_PU_y_

The synthesis route of PBD_x_PU_y_ is illustrated in Figure 1. PBD_x_PU_y_ was synthesized by polycondensation of Pre-PBD and Pre-PU. The average molecular weights of Pre-PBD and Pre-PU, calculated from the ^1^H NMR data using the equations provided in the literature, are 5600 g/mol and 1700 g/mol, respectively [8,33]. The ^1^H NMR spectra of PBD_x_PU_y_ are shown in Figure 2. The integral area of new peaks at 2.64 ppm (b) and 3.48 ppm (d) in Figure 2b, attributed to the hydrogens on an α-carbon atom attached to the carbonyl and amine of the amide bond, respectively [34], is the same. It can be found that with the decrease of Pre-PBD content, the integral area of hydrogen at 4.18 ppm (a) of the α-carbon attached to the oxygen atom of ester decreased and the integral area of hydrogen at 3.24 ppm (c) of the α-carbon attached to nitrogen atom in the urea group increased, indicating that the content of the ester group decreased and the content of the urea group increased. This demonstrates that PBD_x_PU_y_ had been successfully synthesized. However, PBD_x_PU_y_ could not be completely dissolved in TFA. Therefore, it is challenging to calculate the number average molecular weight of PBD_x_PU_y_ using ^1^H NMR data.

The chemical structure of Pre-PBD, Pre-PU and PBD_x_PU_y_ was further studied by FTIR spectra. Figure 3 shows the FTIR spectra of the prepolymer and polyureas. As shown in Figure 3, the stretching vibration peak of the amino group is observed at 3340 cm^−1^, while the stretching vibration peak of methylene group is found in the range of 2920–2854 cm^−1^. The bending vibration peak of the amino group appears at 1554 cm^−1^. Additionally, the asymmetric stretching vibration peak of the C-O-C ether bond is observed at 1165 cm^−1^, and the peak at 1727 cm^−1^ corresponds to the stretching vibration of the carbonyl group in the ester group. The stretching vibration peak of the carbonyl group in the urea group is observed at 1635 cm^−1^. Both ^1^H NMR hydrogen spectra and the FTIR spectra confirmed that the resulting polymers were our target products.

Many papers have shown that the carbonyl area also provides detailed information on hydrogen bonding. To further analyze the hydrogen bonding of polyureas, the carbonyl peaks of PBD_x_PU_y_ were differentiated and fitted using Peak Fit v4.12 software. The maximum error of the fitting was less than 0.5%, and the correlation coefficient was higher than 0.995. The results of fitting curves of FTIR spectra for PBD_x_PU_y_ are presented in Figure 4 and Table 1. As shown in Figure 4, the carbonyl stretching region of PBD_x_PU_y_ is divided into five Gaussian peaks [35]. Peaks at 1628 cm^−1^ and 1653 cm^−1^ are attributed to the hydrogen bonded urea carbonyl groups in the disordered and ordered phase, respectively. The peak at 1691 cm^−1^ is attributed to the free urea carbonyl. The peaks at 1674 cm^−1^ and 1728 cm^−1^ are ascribed to the free and hydrogen-bonded ester carbonyl groups, respectively. The peak percentage areas are listed in Table 1. The proportion of hydrogen-bonded ester carbonyl group in PBD_70%_PU_30%_ is only 4.96%, while it reaches 17.06% for PBD_50%_PU_50%_, indicating the increasing interaction between the soft and hard segments with the increase of PU content. The hydrogen bonding interactions between soft and hard segments could significantly affect the thermal properties, crystallization and mechanical properties of the material, which will be discussed later.

### 3.2. Thermal Properties of PBD_x_PU_y_

Thermal properties are very important for polymer materials when it comes to application. Therefore, the thermal properties of PBD_x_PU_y_ were studied by DSC and the results are shown in Figure 5. Only PBD_50%_PU_50%_ did not show a melting peak of its soft segment, due to its high content of urea groups, which are capable of forming extensive hydrogen bonding with soft segments and reducing the crystallinity of soft segments, making it difficult for the material to fold into a regular crystal. All other PBD_x_PU_y_ samples exhibit two distinct melting peaks, corresponding to the melting of the polyester soft segment and polyurea hard segment in the polymer. It should be noted that hard segment of PBD_50%_PU_50%_ has two melting peaks, while the hard segments of other PBD_x_PU_y_ only have one. This could be due to the fact that a polyurea with a different degree of polymerization of its hard segment was formed for PBD_50%_PU_50%_ due to the self-condensation of the Pre-PU that occurs when the soft segment content is relatively low. The melting point of the hard segment is primarily influenced by two factors: degree of polymerization and hydrogen bonding. A higher degree of polymerization of the macromolecular chain leads to a higher melting point. The strong hydrogen bonding interaction between two segments not only results in decreased crystallinity and folding ability of the hard and soft segments, but also restricts the mobility of both segments. As a result, the melting point of the hard segment decreases. As shown in Table 2, the melting point of the soft and hard segments of PBD_70%_PU_30%_ is higher than that of PBD_80%_PU_20%_, which can be attributed to the higher molar content of the polyurea hard segment of PBD_70%_PU_30%_. The hydrogen bonding of the hard segments of PBD_70%_PU_30%_ is higher than that of PBD_80%_PU_20%_, but the interaction between the soft segment and hard segment for PBD_70%_PU_30%_ is lower than that of PBD_20%_PU_80%_. Thus, the melting points of both the soft segment and hard segment increase. However, with further increases of polyurea hard segment content, the melting points of both the soft segment and hard segment decrease. PBD_60%_PU_40%_ shows lower melting points of the soft and hard segments than those of PBD_70%_PU_30%_, which could be ascribed to the larger amount of hydrogen bonds formed between the soft and hard segments in PBD_60%_PU_40%_, as a higher content of hard segment is introduced. These hydrogen bonds restrict chain folding and decrease the crystallization ability of the soft and hard segments, leading to the decreases of their melting points. Furthermore, the melting peak of the hard segment of PBD_60%_PU_40%_ becomes wider and the area becomes larger as the polyurea content increases, implying a higher crystallinity of the hard segment. This is due to the self-polycondensation of the hard segment of PBD_x_PU_y_. Due to the same reason, different lengths of hard segment have been formed, resulting in the two *T*_m2_ of PBD_50%_PU_50%_. As the melting point of the hard segment is above 180 °C, it can be concluded that the newly synthesized PBD_x_PU_y_ has a relatively high melting point.

DMA is more sensitive to the movement of macromolecular chains, and Figure 6 shows the tan *δ* versus temperature curves of PBD_x_PU_y_. It can be observed from Figure 6 that PBD_x_PU_y_ exhibits two glass transition temperatures (*T*_g_) ranging from −80 °C to 0 °C, suggesting it possesses excellent toughness at low temperatures. For *T*_g_ in the range of −40 °C to 0 °C, the motion of the polyester soft segments is referred to as the *α* transition or micro-Brownian chain motion. Sub-*T*_g_, in the temperature range of −50 °C to −80 °C, is related to the *β* transition, corresponding to the localized chains or small unit motion [36]. For the *β* transition, the peak intensity and width of PBD_50%_PU_50%_ and PBD_60%_PU_40%_ increase compared to PBD_80%_PU_20%_, suggesting that the number of the small units that can mobilize increases. This phenomenon may occur because more of the soft segments in the polyureas cannot crystallize due to hydrogen bonding, which leads to the decreased crystallizability of the polyureas, resulting in the fact that the number of movable groups and units in the amorphous region increases.

### 3.3. Thermal Stability of PBD_x_PU_y_

The thermal stability and *T*_m_ of polymers determine their processability. Figure 7 displays the curves of thermogravimetric analysis (TGA) and derivative thermogravimetry (DTG) for PBD_x_PU_y_. It can be found from Figure 7 that the TGA and DTG curves of PBD_x_PU_y_ exhibit similar shapes, suggesting that the synthesized polymers have similar thermal decomposition mechanisms. Moreover, all PBD_x_PU_y_ have initial decomposition temperatures (*T*_5%_) higher than 316 °C, suggesting that the synthesized PBD_x_PU_y_ samples possess excellent thermal stabilities. The thermal decomposition behaviors of the polymers can be analyzed through the DTG curves, which shows a three-stage degradation process: the initial maximum decomposition rate at 351 °C, which can be interpreted as the breakage of C-N bond in urea group [5], the second stage of decomposition, which its maximum decomposition rate at 415 °C due to the breakage of ester groups [34] and the third maximum decomposition rate occurs at 450 °C, which is caused by the destruction of the long carbon chains. Table 3 shows the main degradation temperatures of PBD_x_PU_y_. All *T*_5%_ (>316 °C) of PBD_x_PU_y_ are evidently higher than the *T*_m_ (198 °C), implying that they have good thermal stability during processing and their processing-temperature range is wide.

### 3.4. Crystallization Properties of PBD_x_PU_y_ Films

The WAXD diffraction pattern was used to further analyze the crystalline properties of PBD_x_PU_y_ and the results are shown in Figure 8. Two distinct peaks with 2θ values at 21.22° and 23.56° can be observed for PBD_80%_PU_20%_ and PBD_70%_PU_30%_, which are almost identical to the diffraction peaks of PBD as previously reported in the literature (2θ = 21.39° and 23.72°) [37]. This suggests that the polyester soft segment in the polymer is highly crystallized. To further analyze the crystallinity of PBD_x_PU_y_, the WAXD peaks of PBD_x_PU_y_ were differentiated and fitted using Peak Fit v4.12 software. The peaks of WAXD spectra of all four samples can be divided into four Gaussian-fitted peaks (21.39°, 23.72°, 20.60°, 22.70°) according to the literature [5,37], as shown in Figure 9. The maximum fitting error was less than 0.5% and the correlation coefficient was all greater than 0.995. The crystallinity was calculated and is listed in Table 4. PBD_50%_PU_50%_ shows a wide and small peak, suggesting a low crystallinity of its soft segment. This could be attributed to its low content and the strong intermolecular hydrogen bonding between the polyurea hard segment and polyester soft segment. This is consistent with the DSC results. For DSC curves, there is no melting peak for the soft segment of PBD_50%_PU_50%_ seen in the DSC. Due to the poor crystallization ability of PBD_50%_PU_50%_, it has an elongation at break as high as 135.0%, which will be discussed later. Both the WAXD and DSC of all samples, except for PBD_70%_PU_30%_, testified that the crystallinity of polyester soft segments reduces with decreasing content of Pre-PBD.

### 3.5. Mechanical Properties of PBD_x_PU_y_

The mechanical properties of polymers are very important for polymer materials when come to applications. Figure 10 displays the tensile curves of PBD_x_PU_y_, while Table 5 summarizes the mechanical property data of PBD_x_PU_y_. It can be found that the tensile strength of PBD_x_PU_y_ increases with the content of Pre-PU. The highest tensile strength of 46.1 MPa is observed when the mass content of Pre-PU is 50%. This can be attributed to the higher content of double-dentate hydrogen bonding between the polyurea hard segment that enhances the tensile strength of the polymer. PBD_80%_PU_20%_ demonstrates the highest elongation at break of 197.8%, which could be explained by the fact that a high Pre-PBD soft segment content endows good toughness of the polyurea. It should be noted that the elongation at break of PBD_50%_PU_50%_ is 135.0%, which is higher than that of PBD_70%_PU_30%_ and PBD_60%_PU_40%_. This is due to fact that the content of Pre-PU is higher in PBD_50%_PU_50%_, which leads to more hydrogen bonding between the soft and hard segments, restricting the macromolecular chain folding and decreasing the crystallization of the soft segment. Previous DSC and WAXD results also indicate the same. It should be noted that the flexural strength, flexural modulus and impact strength of PBD_x_PU_y_ also increase with increasing Pre-PU content. The impact strength of PBD_50%_PU_50%_ can reach as high as 37.2 kJ/m^2^ for PBD_50%_PU_50%_, which is enhanced by about four times as compared to PBD_80%_PU_20%_ or PU-10. This is due to the fact that with the increase of Pre-PU content, the amount of the hydrogen bonding between the hard and soft segments increases substantially and the crystallinity of the soft segment in the PBD_50%_PU_50%_ reduces significantly. This improves the toughness and impact resistance of PBD_50%_PU_50%_. The polymer not only exhibits good toughness, but also possesses high strength and modulus. Therefore, it can be safely concluded that the synthesized polyureas have good combined mechanical properties.

### 3.6. Morphology of PBD_x_PU_y_

The cross-section morphologies of different polymers were observed by SEM to analyze their fracture behaviors. The SEM images of the fracture of PBD_x_PU_y_ are displayed in Figure 11. It can be observed that fracture surface of PBD_x_PU_y_ is rough, which indicates a tough nature. Therefore, PBD_x_PU_y_ possesses high impact resistance, which has been also indicated by the results shown in Table 5. The fracture surfaces of PBD_80%_PU_20%_ appear relatively smooth, while PBD_60%_PU_40%_ and PBD_50%_PU_50%_ display a relatively rough surface with more pits. It should be noted that a myriad of fibrillation and fiber pulling-out phenomena occurred for PBD_50%_PU_50%_, which is the reason for its good toughness.

## 4. Conclusions

In this work, polyester polyureas, named in the format of PBD_x_PU_y_, have been synthesized for the first time using a non-isocyanate route by using polyester as a soft segment. The incorporation of polyester endows toughness upon the polyureas. Hydrogen bonding between the soft and hard segments restricts the macromolecular chain folding of soft segments and decreases the crystallinity of the soft segments. The impact resistance of PBD_50%_PU_50%_ reaches 37.2 kJ/m^2^, which is enhanced by about four times compared to PU-10 and PBD_80%_PU_20%_. Meanwhile, PBD_50%_PU_50%_ has an excellent tensile strength of 46.1 MPa and elongation at break of 135.0%. Additionally, PBD_50%_PU_50%_ shows good thermal properties, with a Tm above 180 °C and *T*_5%_ above 320 °C, which is significantly higher than its processing temperature (210 °C). These polyureas with soft segments prepared from a non-isocyanate route possess excellent impact resistance, tensile strength and thermal properties. They have wide range of potential applications in the field of polymer materials in the future.

## Figures and Tables

**Figure 1 polymers-16-00993-f001:**
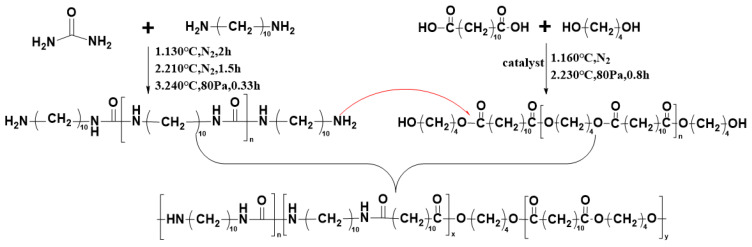
Synthesis route of PBD_x_PU_y_.

**Figure 2 polymers-16-00993-f002:**
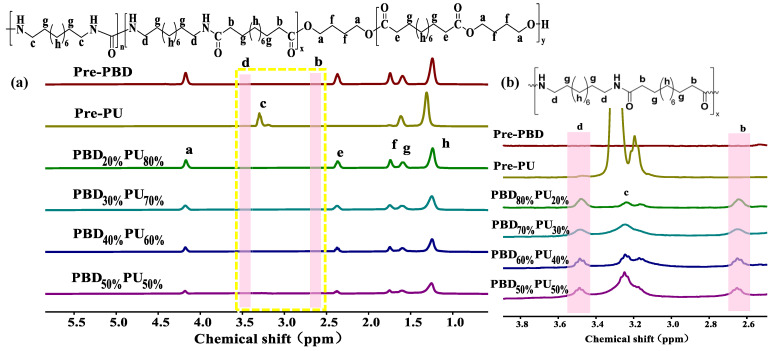
^1^H NMR spectra of PBD_x_PU_y_ (Pre-PBD and Pre-PU): (**a**) the full spectra. (**b**) partially enlarged spectra.

**Figure 3 polymers-16-00993-f003:**
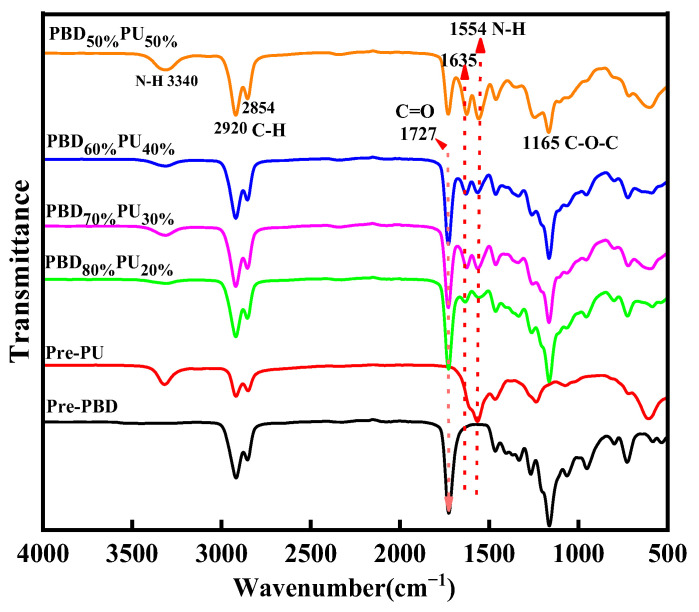
FTIR spectra of Pre-PBD, Pre-PU and PBD_x_PU_y_. The arrows in the diagram are used to indicate the wavenumbers and corresponding function groups.

**Figure 4 polymers-16-00993-f004:**
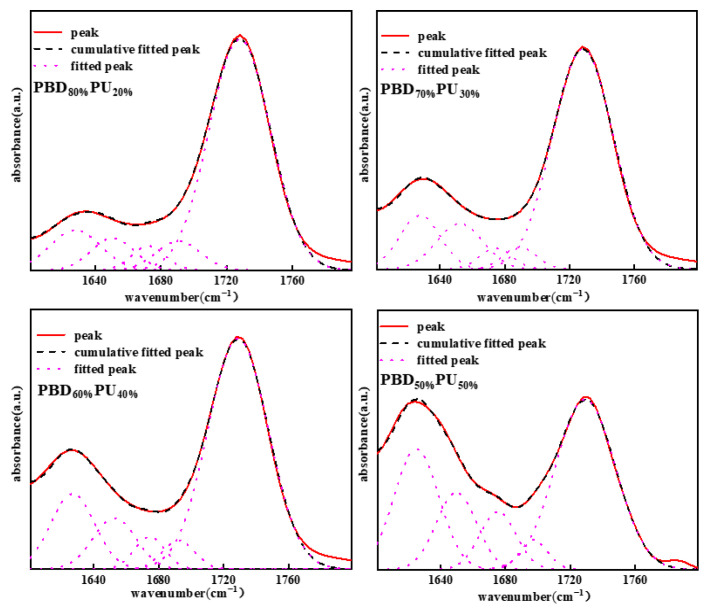
Fitting curves of FTIR spectra of PBD_x_PU_y_.

**Figure 5 polymers-16-00993-f005:**
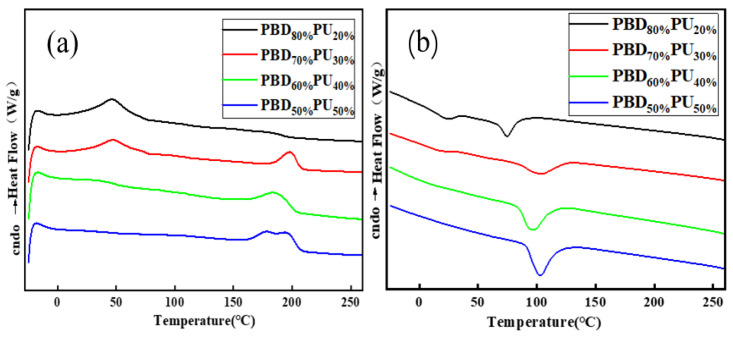
DSC of PBD_x_PU_y_: second heating (**a**) and cooling scans (**b**).

**Figure 6 polymers-16-00993-f006:**
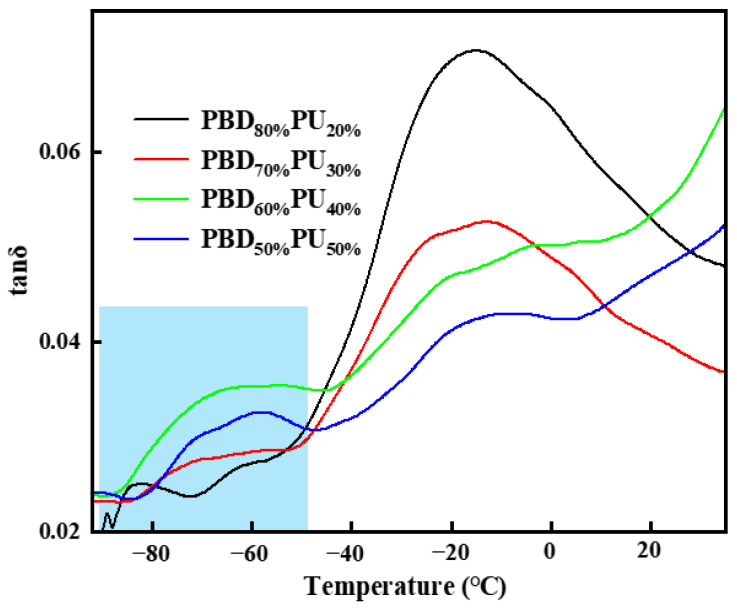
tan δ versus temperature for PBD_x_PU_y_.

**Figure 7 polymers-16-00993-f007:**
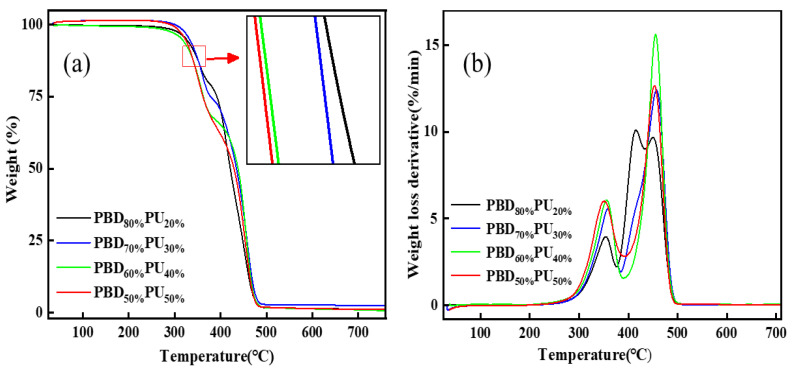
TGA curves (**a**) and DTG curves (**b**) for PBD_x_PU_y_.

**Figure 8 polymers-16-00993-f008:**
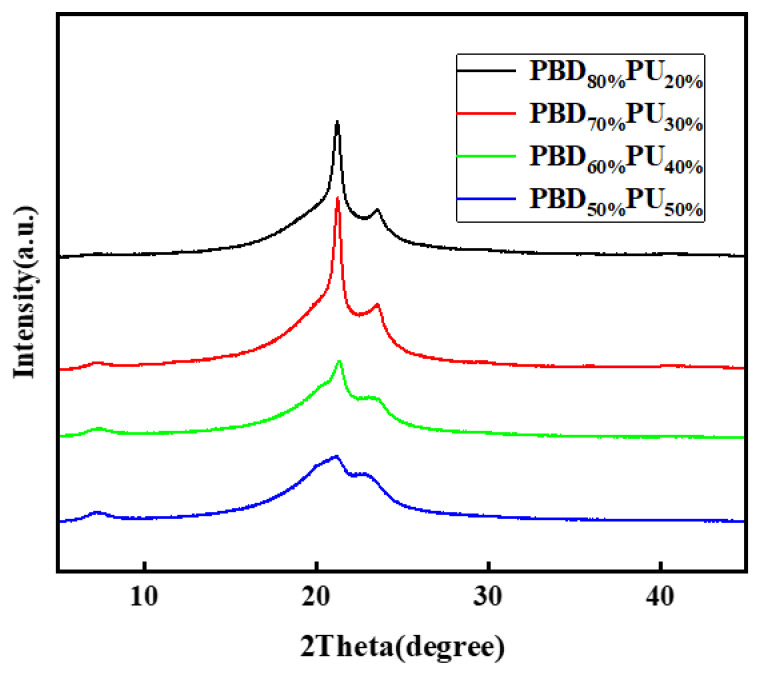
WAXD diffraction pattern of PBD_x_PU_y_ films.

**Figure 9 polymers-16-00993-f009:**
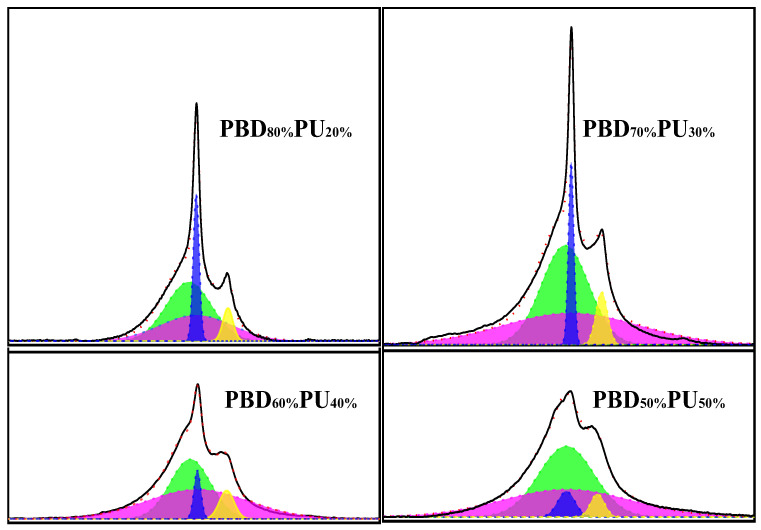
Fitting curves of WAXD diffraction pattern of PBD_x_PU_y_.

**Figure 10 polymers-16-00993-f010:**
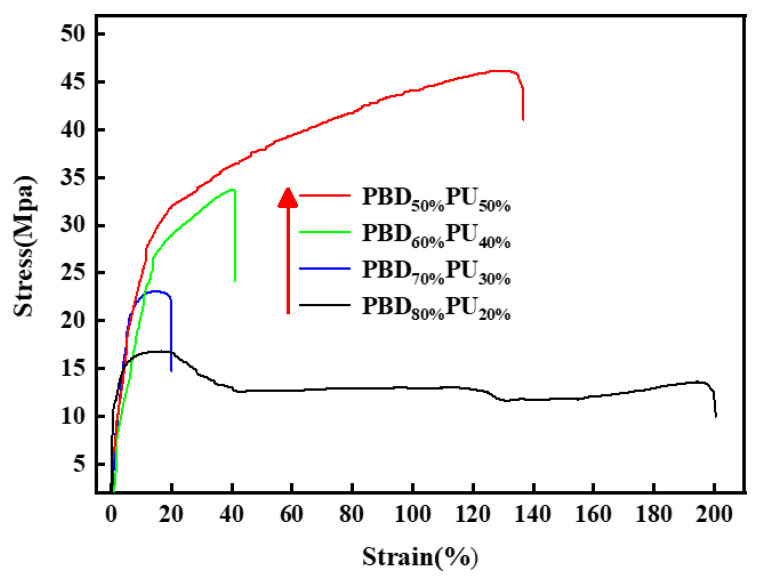
Stress-strain curves of PBD_x_PU_y_.

**Figure 11 polymers-16-00993-f011:**
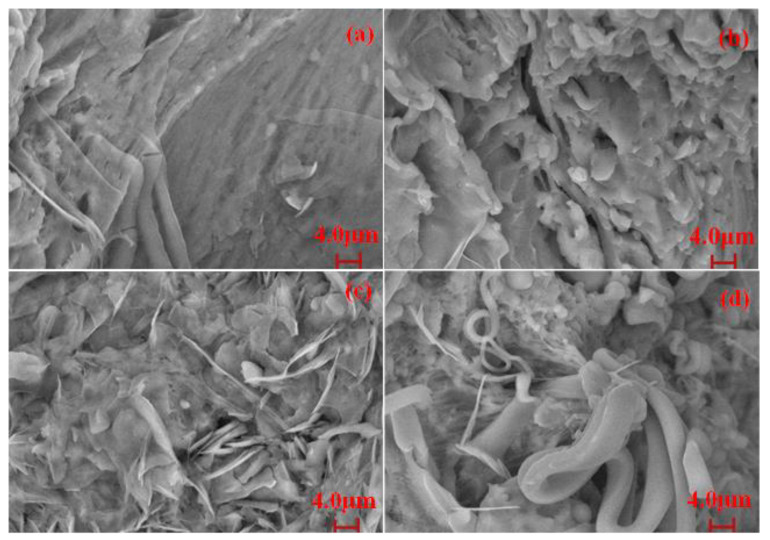
SEM images of quenched section of (**a**) PBD_80%_PU_20%_, (**b**) PBD_70%_PU_30%_, (**c**) PBD_60%_PU_40%_, (**d**) PBD_50%_PU_50%_.

**Table 1 polymers-16-00993-t001:** Calculated relevant parameters from FTIR-fitting results.

	PBD_80%_ PU_20%_	PBD_70%_ PU_30%_	PBD_60%_ PU_40%_	PBD_50%_ PU_50%_
1626–1628 urea (bonded, ordered) A_1_	10.34%	11.51%	15.48%	25.03%
1652–1654 urea (bonded disordered) A_2_	7.00%	11.11%	10.25%	14.34%
1690–1691 urea (free) A_3_	6.54%	3.89%	4.33%	3.76%
1672–1674 ester (bonded) A_4_	4.10%	3.6%	4.63%	9.70%
1728–1729 ester (free) A_5_	72.05%	79.90%	65.31%	47.16%
(A_1_ + A_2_)/(A_1_ + A_2_ + A_3_)	72.57%	85.34%	85.59%	91.28%
A_4_/(A_4_ + A_5_)	5.38%	4.90%	6.61%	17.06%

**Table 2 polymers-16-00993-t002:** Thermal performance parameters of PBD_x_PU_y_.

	*T*_m1_ (°C)	Δ*H*_m_ (J/g)	*T*_m2_ (°C)	Δ*H*_m_ (J/g)	*T*_c1_ (°C)	*T*_c2_ (°C)	*T*_g_ (°C)	
							β	α
PBD_80%_PU_20%_	46.5	25.7	179.3	3.4	24.1	74.5	−60.5	−15.3
PBD_70%_PU_30%_	47.4	17.7	198.2	9.7	16.9	102.8	−64.9	−13.7
PBD_60%_ PU_40%_	36.7	3.3	188.2	17.5	-	95.9	−59.2	−3.6
PBD_50%_PU_50%_	-	-	180.0, 194.7	21.0	-	102.6	−62.7	−11.2

**Table 3 polymers-16-00993-t003:** Main degradation temperatures of PBD_x_PU_y_.

	*T*_5%_ (°C)	*T*_max1_ (°C)	*T*_max2_ (°C)	*T*_max3_ (°C)
PBD_80%_PU_20%_	320	354	415	450
PBD_70%_PU_30%_	316	358	416	457
PBD_60%_PU_40%_	331	356	-	455
PBD_50%_PU_50%_	325	351	-	453

**Table 4 polymers-16-00993-t004:** Crystallinity of PBDxPUy.

Property	PBD_80%_ PU_20%_	PBD_70%_ PU_30%_	PBD_60%_ PU_40%_	PBD_50%_ PU_50%_
Crystallinity (%)	15.70	18.55	12.33	9.29

**Table 5 polymers-16-00993-t005:** Mechanical properties of PBD_x_PU_y_.

Sample	Impact Strength kJ/m^2^	Tensile Strength MPa	Elongation at Breaking%	Flexural Modulus MPa	Flexural Strength MPa
PBD_80%_PU_20%_	7.5	16.5	197.8	269.0	9.8
PBD_70%_PU_30%_	24.5	23.0	18.2	423.1	14.8
PBD_60%_PU_40%_	31.1	33.6	41.4	456.4	15.6
PBD_50%_PU_50%_	37.2	46.1	135.0	524.6	18.6
PBD	30.0	13.0	380.1	500.4	16.2
PU-10	6.8	70.0	24.0	1402.7	48.8

## Data Availability

Data is contained within the article. The original contributions presented in the study are included in the article, further inquiries can be directed to the corresponding author.

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
