# Peer review of "Synthesis, Structure and Properties of Polyester Polyureas via a Non-Isocyanate Route with Good Combined Properties"

_polymers, 2024, doi:10.3390/polym16070993_

Round 1

Reviewer 1 Report

Comments and Suggestions for Authors

The review is attached.

Author Response

Response to Reviewer1Comments

Dear editors and referees,

Thanks for your brilliant advice, we have revised the manuscript very carefully according to your suggestion.

Point 1: Fig 2A is not mentioned in the main text, I suggest A and B figures should be merged as Fig 2, using Fig 2 separately for A and B figures is confusing.

Response1: Thank you for raising the issue. It has been revised as follows and replaces the former one in the manuscript.

Figure 2. 1H NMR spectra of PBDxPUy (Pre-PBD and Pre-PU): (a) the full spectra. (b) partially enlarged spectra.

Point 2: The authors should present the results of prePU-prePBD blend (with no chemical reaction between them) to prove that the last chemical step showed in Fig 1 was successful.

Response 2: Thank you for your significant suggestions. Pre-PBD (40g) and Pre-PU (40g) was fed to a kettle equipped with a nitrogen inlet and a fractionating column and stirred at 230 °C under for 2h. Then the blend product of Pre-PBD50%-Pre-PU50% (Pre-PBD-Pre-PU) was obtained, and its chemical structure was characterized by 1H NMR. The 1H NMR results of Pre-PBD-Pre-PU is showed as follows. It can be observed from Figure 3 that no new peaks appear at 2.64 ppm or 3.48 ppm in Pre-PBD50%-Pre-PU50% sample, suggesting that Pre-PBD did not react with Pre-PU. Compared with the 1H NMR result of PBD50%PU50%, it demonstrates that the last chemical step showed in Fig 1 was successful.

Figure 3. 1H NMR spectra of PBD50%PU50% and Pre-PBD50%-Pre-PU50%

Point 3: The crystallinity of PBD70%PU30% sample is unexpectedly higher; the authors should suggest something for this phenomenon. Furthermore, in row 331 (page 10) the name of this sample is wrong (the numbers were swapped).

Response 3: Thanks for your kindly suggestions. The reason for the highest crystallinity of the soft segments of PBD70%PU30% is unclear within our knowledge. The reason for this phenomenon may be attributed to the thermal history of the soft segments of the PBD70%PU30% or the test condition, which results in the highest crystallinity.

Thank you for raising the issue. The name of the sample has been corrected in the manuscript.

Point 4: In the conclusion section there is 46.5 MPa, but in table 5 it is 46.1 MPa.

Response 4: It has been revised according to your kind suggestion.

Please don't hesitate to contact me if you have any questions.

Best regards

Sincerely yours

Liuchun Zheng

Professor

School of Chemical Engineering and Technology, Xinjiang University

School of Chemical Engineering and Technology & State Key Laboratory of Separation Membranes and Membrane Processes, Tiangong University

Tel/Fax: +86 13910436604

E-mail: liuchunzheng@tiangong.edu.cn; hubeizlc@126.com

Reviewer 2 Report

Comments and Suggestions for Authors

In this manuscript, polyester polyurea through a green route was synthesized, and characterized for mechanical, thermal and structural properties. Synthesized polyester-polyurea samples exhibit good mechanical as well as thermal properties. This work is well organized, logically addressed, and high quality presented. I would suggest accept after minor issues:

1. For XRD, please include the source and wavelength in the experimental part. Otherwise, 2theta figures are meaningless. Or change Fig. 8 x axis to q also works.

2. Please specify how modulus was calculated (e.g. 0 to 1% or 5%)

3. For SEM samples, are the samples coated with gold? There is barely charging observed, however it didn't mention sputtering a thin layer of gold in the experimental description.

4. There are two Fig. 2s. Please recheck figure numbers.

5. In Table 4, why 70-30 has the highest crystallinity? The crystallinity from DSC cannot be calculated. Is the heat of fusion for the two polymers reported before?

Author Response

Response to Reviewer 2 Comments

Dear editors and referees,

Thanks for your brilliant advice, we have revised the manuscript very carefully according to your suggestion.

Point 1: 1. For XRD, please include the source and wavelength in the experimental part. Otherwise, 2theta figures are meaningless. Or change Fig. 8 x axis to q also works.

Response 1: Thank you for your advice, we have reviewed it as below.

“Wide angle X-ray analysis (WAXD) was determined using an X-ray diffractometer (Bruker D8 DISCOVER). The samples were scanned from 7° to 45° at a scanning speed of 2°/min using a Cu Kα radiation (λ = 154 nm) target.”

Point 2: Please specify how modulus was calculated (e.g. 0 to 1% or 5%)

Response 2: Thank you for your professional question, we have checked the original data, and review it as below.

 “The bending modulus was calculated from the slope of the stress-strain curve in the strain range of 0~1%."

Point 3: For SEM samples, are the samples coated with gold? There is barely charging observed, however it didn't mention sputtering a thin layer of gold in the experimental description.

Response 3: Thank you for your professional question, which is very important to improve our manuscript. “The fracture surfaces of samples were coated with gold before they were characterized using a scanning electron microscope (MERLIN Compact, Zeiss) with an accelerating voltage of 15 kV.”

Point 4: There are two Fig. 2s. Please recheck figure numbers.

Response 4: Thank you for raising the issue. It has been revised as follows and replaces the former one in the manuscript.

Figure 2. 1H NMR spectra of PBDxPUy (Pre-PBD and Pre-PU): (a) the full spectra. (b) partially enlarged spectra.

Point 5: In Table 4, why 70-30 has the highest crystallinity? The crystallinity from DSC cannot be calculated. Is the heat of fusion for the two polymers reported before?

Response 5: Thank you for your careful and professional work on our manuscript. The crystallinity of PBD30%PU30% was challenging to calculate using DSC result because the ΔHmθ values of PBD and PU were not reported before. The reason for the highest crystallinity of the soft segments of PBD70%PU30% is very complicated and unclear within our knowledge. The reason for this phenomenon may be attributed to the thermal history of the soft segments of the PBD70%PU30% or the test condition, which results in the highest crystallinity.

Please don't hesitate to contact me if you have any questions.

Best regards

Sincerely yours

Liuchun Zheng

Professor

School of Chemical Engineering and Technology, Xinjiang University

School of Chemical Engineering and Technology & State Key Laboratory of Separation Membranes and Membrane Processes, Tiangong University

Tel/Fax: +86 13910436604

E-mail: liuchunzheng@tiangong.edu.cn; hubeizlc@126.com
